# Robust prognostic prediction model developed with integrated biological markers for acute myocardial infarction

**Masahiro Nishi** [1]*, **Eiichiro Uchino**[2], **Yasushi Okuno**[2], **Satoaki Matoba**[1]

**1** Department of Cardiovascular Medicine, Graduate School of Medical Science, Kyoto Prefectural University of Medicine, Kyoto, Japan, **2** Department of Biomedical Data Intelligence, Graduate School of Medicine, Kyoto University, Kyoto, Japan

\* nishim@koto.kpu-m.ac.jp

**Data Availability Statement:** All relevant data are within the manuscript and its Supporting Information files.

## Abstract

Commonly used prediction methods for acute myocardial infarction (AMI) were created before contemporary percutaneous coronary intervention was recognized as the primary therapy. Although several studies have used machine learning techniques for prognostic prediction of patients with AMI, its clinical application has not been achieved. Here, we developed an online application tool using a machine learning model to predict in-hospital mortality in patients with AMI. A total of 2,553 cases of ST-elevation AMI were assigned to 80% training subset for cross validation and 20% test subset for model performance evaluation. We implemented random forest classifier for the binary classification of in-hospital mortality. The selected best feature set consisted of ten clinical and biological markers including max creatine phosphokinase, hemoglobin, heart rate, creatinine, systolic blood pressure, blood sugar, age, Killip class, white blood cells, and c-reactive protein. Our model achieved high performance: the area under the curve of the receiver operating characteristic curve for the test subset, 0.95: sensitivity, 0.89: specificity, 0.91: precision, 0.43: accuracy, 0.91 respectively, which outperformed common scoring methods. The freely available application tool for prognostic prediction can contribute to risk triage and decision-making in patient-centered modern clinical practice for AMI.

## Introduction

Myocardial infarction is still a major critical disease with high mortality and morbidity, although coronary reperfusion therapy has improved the prognosis [1]. Several risk scoring methods have been proposed to predict short-and long-term mortality in patients with acute myocardial infarction (AMI) and have described different biological predictive factors to date [2–7]. Commonly used scoring methods, such as the TIMI risk index and GRACE score, were created before contemporary percutaneous coronary intervention (PCI) was recognized as a fundamental therapy. In addition, other proposed models involve impractical issues owing to their inaccuracy or cumbersome method. Accurate prognostic prediction model in modern

**Funding:** Satoaki Matoba was supported by the JST COI Grant Number JPMJCE1302. The funders had no role in study design, data collection and analysis, decision to publish, or preparation of the manuscript.

**Competing interests:** The authors have declared that no competing interests exist.

clinical practice for AMI is crucial, given its higher mortality compared to other common diseases.

Machine learning (ML) techniques in medical science have been shown to successfully overcome the limitations of predictors that rely on a single feature by combining different types of features even in a nonlinear fashion [8, 9], because none of the predictive factors act in isolation and therefore, cannot independently produce precise prediction. ML has the potential to improve the predictive accuracy for the prognosis of cardiovascular disease [10, 11]. In-hospital death caused by the complication of AMI, such as life-threatening arrhythmia, myocardial rupture, and severe heart failure, is difficult to precisely predict by clinicians. Several studies have used ML techniques for prognostic prediction of patients with myocardial infarction [12–15]; however, its clinical application has not been achieved due to its low versatility and feasibility. In the present study, we developed a state-of-the-art application tool for a prognostic prediction model with integrated biological markers for in-hospital mortality in patients with AMI.

## Methods

### Patient data description

The AMI-Kyoto Multi-Center Risk Study, a large multicenter observational study in which 23 collaborating hospitals in Japan, collected demographic, clinical background, laboratory, procedural, angiographic, and outcome data of patients with AMI. It was established in 2000 to analyze the data and create an emergency hospital network for heart diseases in Kansai district in Japan [16–18]. Indication for PCI and pharmacological approaches are similar because all the hospitals are registered as training facility of Japanese Circulation Society and conduct the guideline-based standard management. Data were collected on admission except for prognosis, max creatine phosphokinase (CPK), and procedural findings. CPK was measured on admission and every 6 h after PCI until it peaked. A total of 3,497 consecutive patients with AMI who were admitted at the AMI Kyoto Multi-Center Risk Study Group Hospitals from January 2009 to December 2015 were enrolled in the study. Power calculation was not performed, given that machine learning-based prediction does not evaluate statistical significance. Cases with out-of-hospital cardiac arrest, non-ST-elevation AMI (STEMI), without coronary angiography (CAG), and missing prognosis were excluded, and 2,553 cases were analyzed.

After informed consent to participate in the AMI-Kyoto Multi-Center Risk Study was confirmed by each patient, all in-hospital data were transmitted to the center located at the Department of Cardiology in Kyoto Prefectural University of Medicine for the analysis. Because this study corresponds to an observational study that uses only existing data from medical records, we provided verbal informed consent without the agreement document. We provided a record of the informed consent to the electronic medical recording system. After each institution approved participation in the study, the study was approved by the ethics committee of Kyoto Prefectural University of Medicine as a general organization (approval number ERB-C-1865). The study conformed to the principles outlined in the Declaration of Helsinki.

### Model description

In-hospital mortality was set as the target, and a total of 32 clinical and biological markers were set as the features of prediction modelling. We implemented random forest classifier (RF) for the binary classification of in-hospital mortality using the scikit-learn 0.24.1 package [19] in Python 3.7.4. We split the dataset using the *train_test_split* function which randomly partitions a dataset into training and test subsets with test size = 0.2, to generate the training

(80%) and test (20%) datasets. Model performance was evaluated using both the training and test subsets. The *RFE* (recursive feature elimination) function was used for the best feature selection in the training dataset, where RF was deployed as an estimator. As a result, ten features, such as max CPK, hemoglobin (Hb), heart rate (HR), creatinine (Cr), systolic blood pressure (BPs), blood sugar (BS), age, Killip class, white blood cells (WBC), and c-reactive protein (CRP), were selected for the best feature set. The feature importance for all features and the selected ten features were calculated by mean decrease impurity method using *feature_importances_* function. For tuning RF with the best hyperparameters, we used *the GridSearchCV* function to the training subset with 5-fold cross validation. A total of 960 RF models were evaluated with different combinations of n_estimators, max_depth, and min_samples_split hyperparameters. Consequently, we selected a model with n_estimators = 200, max_depth = 6, and min_samples_split = 5 hyperparameters which showed the highest area under the curve (AUC) = 0.90. Using these hyperparameters, the final model was trained with the whole training subset. The KOTOMI risk calculator is available in the Heroku web application platform (https://kotomi-calculator.herokuapp.com).

We visualized the receiver operating characteristic (ROC) and precision-recall curves and calculated the AUC for the test subset using the Precrec package [20] in R3.6.1 [21]. We determined the optimal threshold of the model, which discriminates between death and survival using Youden's index method using the pROC package [22]. The confusion matrix, F1-score, sensitivity, specificity, precision, and accuracy were also evaluated to compare the model performance with the TIMI risk index and GRACE score.

## Results

We obtained sample data of 2,553 patients with STEMI undergoing CAG from a total of 3,497 participants (S1 Fig and S1 Appendix). Data were collected on admission except for prognosis, CPK, and procedural findings (Table 1). Of these, 2,358 survived and 195 died during hospitalization. Median (IQR) age for the survivors and deceased group was 69 (61–78) and 81 (73–86); body mass index (BMI) was 23.1 (20.9–25.4) and 22.0 (19.4–24.7); BPs was 132 (112–154) and 106 (80–134); HR was 73 (60–88) and 83 (61–105); WBC was 9500 (7600–11900) and 10320 (7855–13915); Hb was 14.1 (12.7–15.4) and 12.2 (10.5–14.0); BS was 153 (128–199) and 201 (147–279); max CPK was 1814 (832–3376) and 3350 (1346–6924); Cr was 0.8 (0.7–1.0) and 1.2 (0.9–1.6); CRP was 0.2 (0.1–0.5) and 0.7 (0.1–3.8); GRACE score was 158 (137–180) and 219 (194–241); and TIMI risk index was 25.6 (18.3–35.2) and 47.8 (33.9–65.2). Characteristics in the survival and deceased groups were female: 25.7% vs 45.1%; hypertension: 61.9% vs 69.1%; diabetes: 30.8% vs 35.4%; dyslipidemia: 44.1% vs 34.9%; smoking: 41.8% vs 21.1%, respectively. Other characteristics are shown in Table 1.

An overview of the application development of the prognostic prediction model in clinical practice for patients with STEMI is presented in Fig 1. A total of 2,553 cases were randomly divided into training (80%) and test (20%) subsets with similar characteristics (S1 Table). The feature importance for all features was calculated using the RF as a model estimator (Fig 2A). Most of the highly important features were composed of basic clinical and biological markers such as vital signs and common blood markers, whereas the findings of CAG and PCI or emergency system-dependent factors exhibited a lower impact on in-hospital mortality. The AUC for each feature number was evaluated for training and test subsets in the RF, XGB classifier, and logistic regression (Fig 2B). Eventually, ten features (max CPK, Hb, HR, Cr, BPs, BS, age, Killip class, WBC, and CRP) were selected for the feature set to establish the RF model because of its high adaptability for the training subset (Fig 2B and 2C). Thus, an easily accessible online application tool was developed to calculate the in-hospital mortality risk in the patients with

**Table 1. Characteristics of participants.**

| Characteristic | Survival (n = 2,358) | Dead (n = 195) |
|---|---|---|
| Age, median (IQR) | 69 (61–78) | 81 (73–86) |
| Gender female, n (%) | 596 (25.7) | 106 (45.1) |
| BMI, median (IQR) | 23.1 (20.9–25.4) | 22.0 (19.4–24.7) |
| Hypertension, n (%) | 1424 (61.9) | 121 (69.1) |
| Diabetes, n (%) | 707 (30.8) | 62 (35.4) |
| Dyslipidaemia, n (%) | 1014 (44.1) | 61 (34.9) |
| Smoking, n (%) | 962 (41.8) | 37 (21.1) |
| Family history, n (%) | 170 (7.4) | 1 (0.6) |
| MI history, n (%) | 198 (8.6) | 33 (17.8) |
| CVD history, n (%) | 467 (20.4) | 70 (36.6) |
| HD, n (%) | 30 (1.3) | 25 (12.8) |
| Killip, n (%) | | |
| 1 | 1591 (73.3) | 38 (21.3) |
| 2 | 307 (14.1) | 26 (14.6) |
| 3 | 109 (5.0) | 26 (14.6) |
| 4 | 165 (7.6) | 88 (49.4) |
| BPs, median (IQR) | 132 (112–154) | 106 (80–134) |
| HR, median (IQR) | 73 (60–88) | 83 (61–105) |
| WBC, median (IQR) | 9500 (7600–11900) | 10320 (7855–13915) |
| Hb, median (IQR) | 14.1 (12.7–15.4) | 12.2 (10.5–14.0) |
| BS, median (IQR) | 153 (128–199) | 201 (147–279) |
| max CPK, median (IQR) | 1814 (832–3376) | 3350 (1346–6924) |
| Cr, median (IQR) | 0.8 (0.7–1.0) | 1.2 (0.9–1.6) |
| CRP, median (IQR) | 0.2 (0.1–0.5) | 0.7 (0.1–3.8) |
| TIMI pre, n (%) | | |
| 0 | 1373 (58.4) | 140 (72.5) |
| 1 | 258 (11.0) | 20 (10.4) |
| 2 | 412 (17.5) | 21 (10.9) |
| 3 | 309 (13.1) | 12 (6.2) |
| TIMI post, n (%) | | |
| 0 | 33 (1.4) | 14 (7.4) |
| 1 | 22 (0.9) | 10 (5.3) |
| 2 | 119 (5.1) | 24 (12.6) |
| 3 | 2160 (92.5) | 142 (74.7) |
| Stent, n (%) | 2072 (89.1) | 150 (78.5) |
| Thrombus aspiration, n (%) | 1596 (68.6) | 115 (60.2) |
| Part of MI, n (%) | | |
| Anterior | 1172 (49.9) | 119 (62.3) |
| Posterior-inferior | 998 (42.5) | 58 (30.4) |
| Lateral | 177 (7.5) | 14 (7.3) |
| Culprit vessel, n (%) | | |
| RCA | 934 (40.0) | 52 (26.9) |
| LAD | 1134 (48.6) | 89 (46.1) |
| LMT | 36 (1.5) | 25 (13.0) |
| LCX | 231 (9.9) | 27 (14.0) |
| Numbers of lesion, n (%) | | |
| 1 | 1423 (61.5) | 100 (52.1) |

(*Continued*)

**Table 1.** (Continued)

| Characteristic | Survival (n = 2,358) | Dead (n = 195) |
|---|---|---|
| 2 | 605 (26.2) | 49 (25.5) |
| 3 | 284 (12.3) | 43 (22.4) |
| Season, n (%) | | |
| Spring | 603 (25.6) | 56 (28.7) |
| Summer | 539 (22.9) | 38 (19.5) |
| Autumn | 561 (23.8) | 45 (23.1) |
| Winter | 655 (27.8) | 56 (28.7) |
| Onset time, n (%) | | |
| Diurnal | 1168 (56.6) | 79 (54.9) |
| Nocturnal | 894 (43.4) | 65 (45.1) |
| Door to balloon time, n (%) | | |
| ≦ 1.5 h | 79 (3.5) | 5 (2.9) |
| 1.5 h < | 2182 (96.5) | 167 (97.1) |
| Onset to door time, n (%) | | |
| ≦ 24 h | 2038 (88.2) | 144 (76.6) |
| 24 h < | 272 (11.8) | 44 (23.4) |
| Transfer, n (%) | | |
| Ambulance | 1105 (48.5) | 100 (53.5) |
| Walk-in | 629 (27.6) | 19 (10.2) |
| In-hospital | 67 (2.9) | 17 (9.1) |
| From other hospital | 478 (21.0) | 51 (27.3) |
| Grace Score, median (IQR) | 158 (137–180) | 219 (194–241) |
| TIMI risk index, median (IQR) | 25.6 (18.3–35.2) | 47.8 (33.9–65.2) |

IQR, interquartile range; BMI, body mass index; MI, myocardial infarction; CVD, cardiovascular disease; HD, hemodialysis; BPs, systolic blood pressure; HR, heart rate; WBC, white blood cells; Hb, hemoglobin; BS, blood sugar; CPK, creatine phosphokinase; Cr, creatinine; CRP, c-reactive protein; TIMI, thrombolysis in myocardial infarction; RCA, right coronary artery; LAD, left anterior descending artery; LMT, left main trunk; LCX, left circumflex.

STEMI according to the programming code for the prediction model, then we called it "KOTOMI" after AMI-Kyoto Multi-Center Risk Study (Fig 1 and S2 Fig). The KOTOMI risk calculator is available in a web application platform (https://kotomi-calculator.herokuapp.com).

To evaluate the model performance of KOTOMI, we compared various performance metrics of KOTOMI with common risk scoring methods such as the GRACE score and TIMI risk index on the test subset. The AUC of the ROC curve and area under the precision-recall curve (AUPRC) were calculated. The AUCs of KOTOMI, GRACE, and TIMI were 0.95, 0.88, and 0.84, respectively (Fig 3A). The AUPRC of the KOTOMI, GRACE, and TIMI models were 0.66, 0.54, and 0.36, respectively (S3 Fig). The confusion matrix of each model was determined by the optimal threshold of the model, which discriminates between death and survival by Youden's index method for the ROC curve (Fig 3B). The sensitivities of KOTOMI, GRACE, and TIMI were 0.89, 0,89, and 0.78; specificity was 0.91, 0.81, and 0.79; precision was 0.43, 0.27, and 0.23; F1-Score was 0.58, 0.42, and 0.35; accuracy was 0.91, 0.82, and 0.79, respectively (Fig 3C). Taken together, the KOTOMI model outperformed the commonly used risk scoring methods. Calibration plots showed that high fractions of positives tend to have lower probability than perfect calibration. In the present study, we used the original model for risk calculation (S4 Fig).

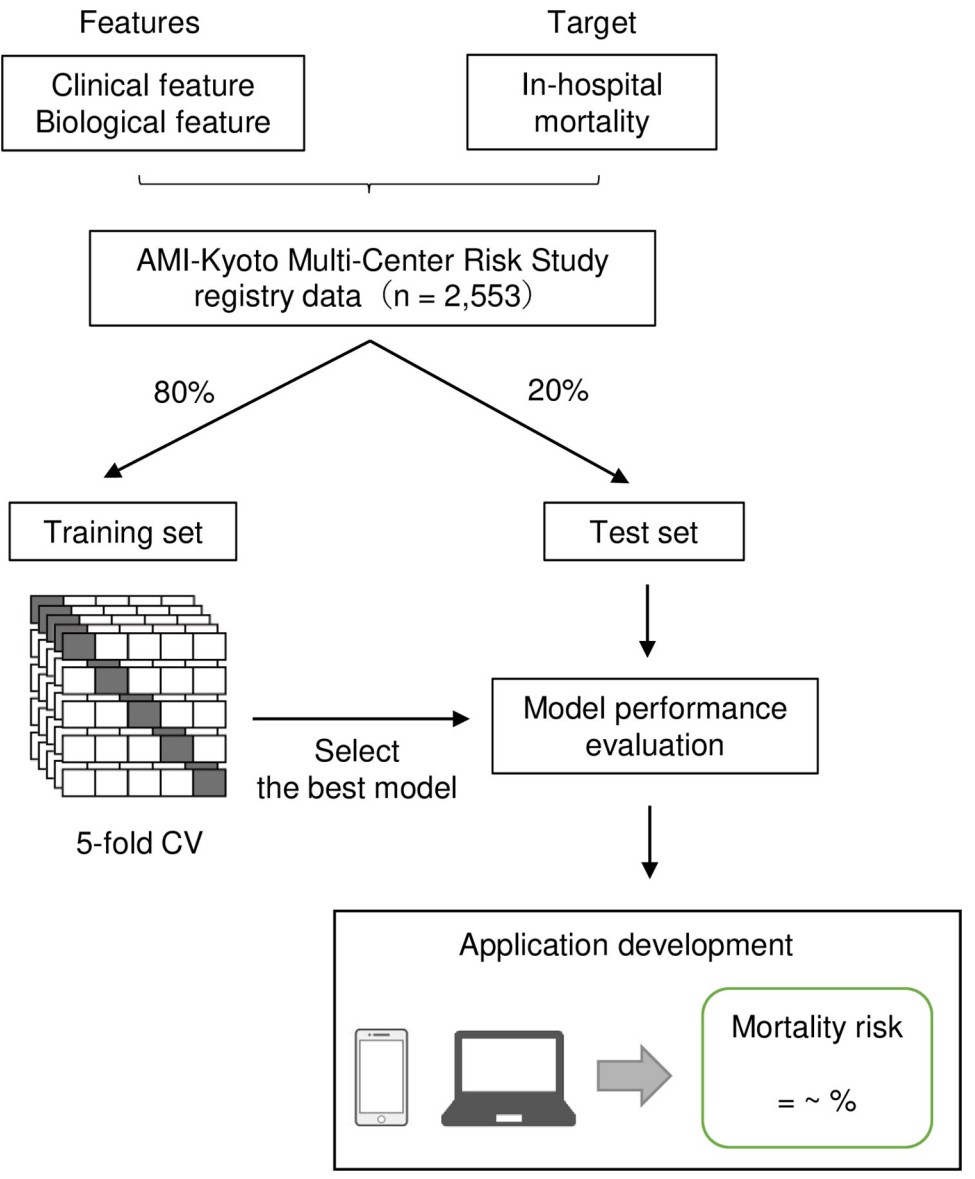

**Fig 1. Overview of development of prognostic prediction model for acute myocardial infarction.** A total of 2,553 cases were divided into training (80%) and test (20%) subsets. A random forest classifier was trained on several clinical and biological features on the training subset using 5-fold cross validation (CV) to predict in-hospital mortality in patients with ST-elevation acute myocardial infarction (STEMI). The resulting trained model with the best hyperparameters was evaluated using various performance metrics using the test subset. The online application tool to predict in-hospital mortality risk in patients with STEMI was developed according to the programming code of the model.

## Discussion

In the present study, we developed a useful tool in which the KOTOMI model was applied to calculate the in-hospital mortality risk in patients with STEMI. The KOTOMI model with high accuracy and performance outperformed common risk scoring methods, such as the TIMI risk index and GRACE score. The development of a precise prediction model for AMI with ML technique could have a significant impact on modern clinical practice for AMI because it can contribute to decision-making in patient-centered medicine.

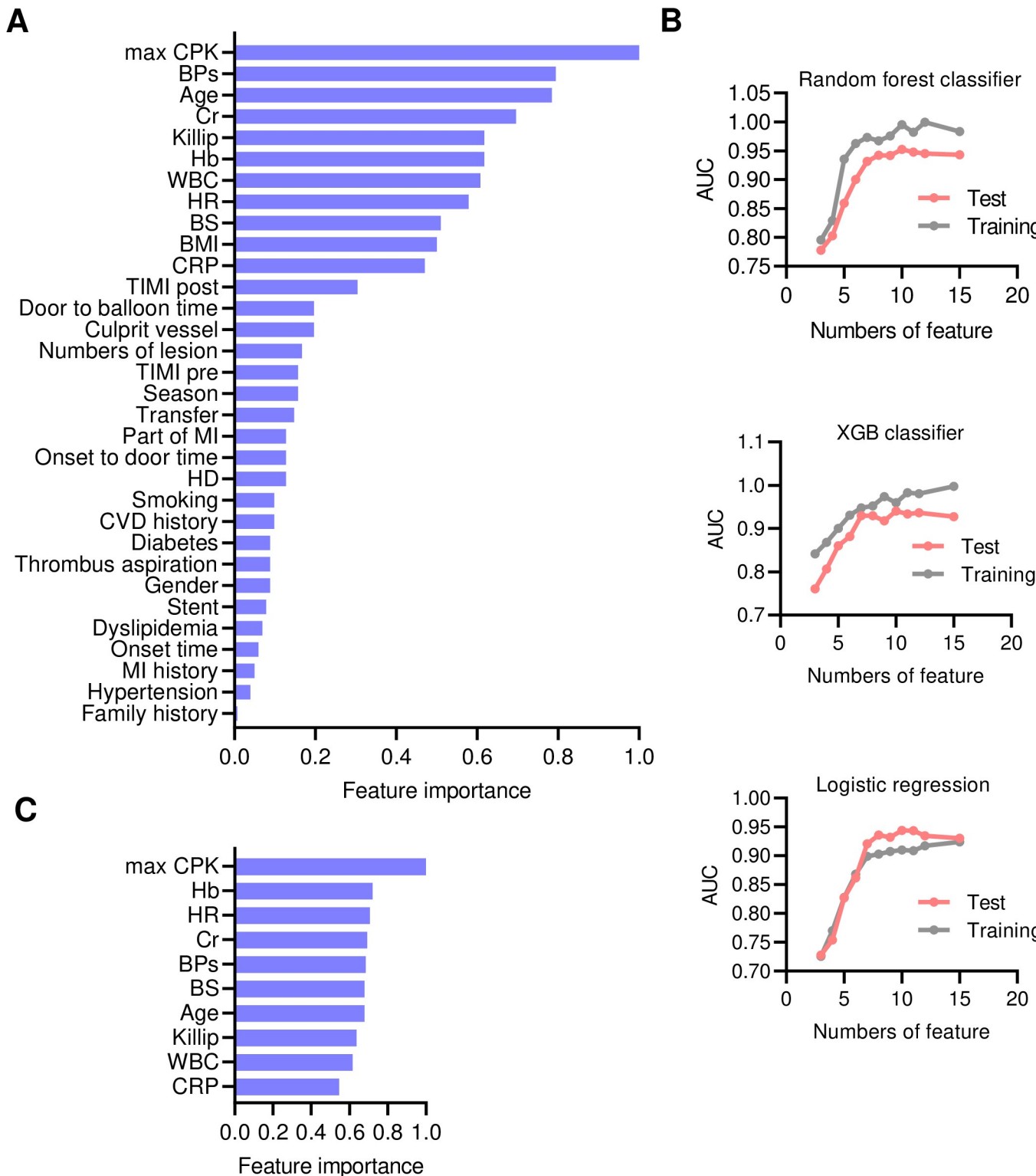

**Fig 2. Feature selection of prognostic prediction model for acute myocardial infarction. A,** Feature importance for all features. **B,** Area under the curve (AUC) for each feature numbers for training and test subsets in random forest classifier, XGB classifier, and logistic regression. **C,** Feature importance for selected best feature set.

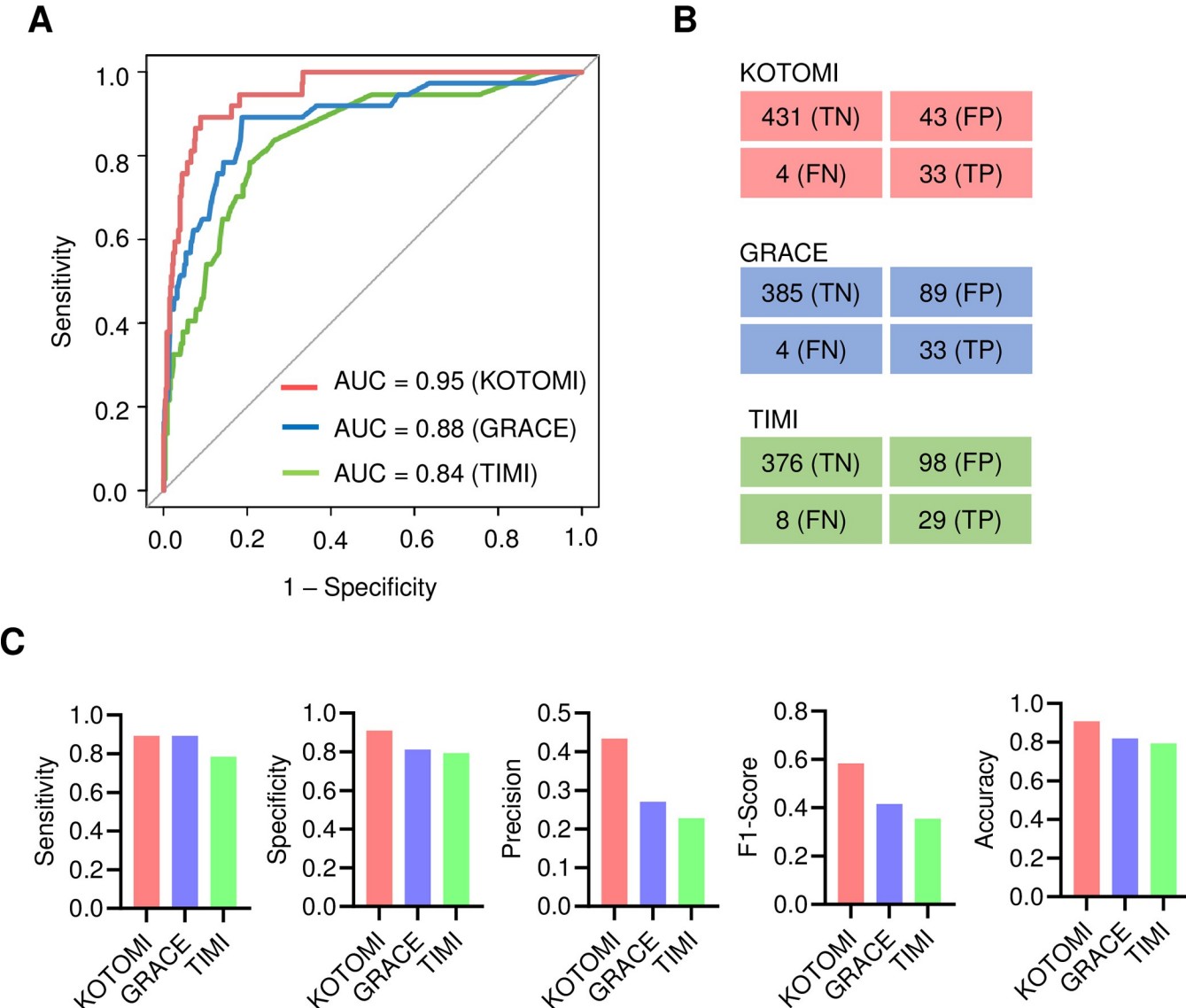

**Fig 3. Model performance evaluation for test subset. A,** Receiver operating characteristic (ROC) curve and its area under the curve (AUC) of KOTOMI, GRACE, and TIMI. **B,** Confusion matrix of each model was determined by the optimal threshold of the model, which discriminates death and survival by Youden's index method for the ROC curve. **C,** Sensitivity, specificity, precision, F1-Score, and accuracy for KOTOMI, GRACE, and TIMI were calculated based on the confusion matrix.

We selected basic clinical and biological markers as the feature set for prognostic prediction. The most important feature was the max CPK in all features. Max CPK is a prognostic predictor for AMI and reflects infarct size [23]. Reperfusion therapy is known to increase the absolute max CPK and quicken its peak time earlier than 24 h after the onset [24]. CPK needs to be monitored until peak out after PCI; thereafter, the KOTOMI model can be applied for prognostic prediction. Hb was also an important feature. Indeed, anemia is a prognostic predictor for AMI [25], which can be improved in daily clinical practice, although the optimal transfusion strategy for anemia in patients with AMI is undefined [26, 27]. Imaging data, such as echocardiogram or cardiac magnetic resonance, may be used as additional features for the prediction. Left ventricular ejection fraction (LVEF) was removed from the patient's

characteristics because of the large number of missing values. LVEF is a powerful individual predictive factor for AMI [28, 29], whereas adding LVEF to the GRACE score did not convey significant prognostic information [30]. The selected features may need to be updated in the future using the contemporary data according to the progress of the standard management for acute myocardial infarction, followed by the update of its application tool. Interestingly, the technical outcomes of catheterization therapy and emergency medical service-related factors had a lower impact than the selected features for the prediction model. This suggests that researchers should focus on primary prevention and amelioration of cardiac damage at the disease onset to improve the prognosis of AMI, given that catheterization techniques and emergency systems have been improved.

The KOTOMI model outperformed the commonly used risk scoring methods. The TIMI risk index is a widely used simple predictive score for short-and long-term mortality in patients with STEMI using three factors: age, HR, and BPs [5, 31]. Subdividing the TIMI risk index by laboratory blood examination, including WBC, Hb, CRP, Cr, and BS levels, reinforces its accuracy [16]. However, the risk score was created among patients treated with thrombolytic therapy before PCI was recognized as a primary therapy for AMI. The GRACE score uses eight risk factors including age, Killip class, BPs, ST-segment deviation, cardiac arrest during presentation, Cr, positive initial cardiac enzyme findings, and HR. It predicts in-hospital death and six-month death rate in patients with acute coronary syndrome [6, 32]. The GRACE score maintains durable performance in contemporary practice for AMI [30, 33]. Other risk scores such as PAMI [34], Zwolle [35], and CADILLAC [28] have been proposed to date. To our knowledge, these scoring methods are not commonly used compared with the TIMI risk index and GRACE score because of their inaccuracy or cumbersome method. Several studies have used ML techniques for prognostic prediction of AMI [12–15]; however, their clinical application has not been achieved due to its low versatility and feasibility.

The present study has several limitations. The KOTOMI model was created using Japanese participants. Therefore, the model performance comparing with TIMI risk index and GRACE will need to be evaluated in larger cohorts of various races. The model was created using the data between January 2009 to December 2015. Validation for current data will be necessary for further study. To enhance the model's applicability, external validation should be performed. The present study was designed to predict only in-hospital mortality, given that the nature of the myocardial infarction incurs critical consequences in the short term and the long-term prognosis implies other chronic disease conditions. Troponin, which is a standard biomarker for diagnosis of STEMI and associated with its severity, CK-MB, BNP and NT-proBNP were not measured in the study.

In conclusion, we developed a state-of-the-art tool for prognostic prediction using the ML technique in patients with STEMI. The developed KOTOMI model can contribute to risk triage and decision-making in patient-centered modern clinical practice for AMI.

## Supporting information

**S1 Table. Characteristics of patients in training and test subsets.**
(DOCX)

**S1 Fig. Flow diagram of sample collection and filtering process.**
(DOCX)

**S2 Fig. KOTOMI risk calculator for in-hospital mortality risk in patients with acute myocardial infarction.** Online application tool of KOTOMI model to predict in-hospital mortality risk in patients with ST-elevation acute myocardial infarction (STEMI) was developed in

Heroku web application platform according to the programming code of the model. An application programming interface (API) that returns the prediction results was tested to show the correct behavior for a subset of records in the dataset.
(DOCX)

**S3 Fig. Model performance illustrated by precision-recall curve for test subset in KOTOMI, GRACE score, and TIMI risk index.** AUPRC indicates area under precision-recall curve.
(DOCX)

**S4 Fig. Model calibration.** Calibration plots were depicted for original model (left panel) and model calibrated by Isotonic Regression (right panel) using training data (A) and test data (B). Samples were divided into ten bins.
(DOCX)

**S1 Appendix. Data set for study analysis.**
(XLSX)

## Acknowledgments

The authors thank all the members of AMI-Kyoto Multi-Center Risk Study Group for collecting data.

## Author Contributions

**Conceptualization:** Masahiro Nishi, Satoaki Matoba.

**Data curation:** Masahiro Nishi.

**Formal analysis:** Masahiro Nishi, Eiichiro Uchino, Yasushi Okuno.

**Investigation:** Masahiro Nishi, Satoaki Matoba.

**Supervision:** Satoaki Matoba.

**Writing – original draft:** Masahiro Nishi.

**Writing – review & editing:** Eiichiro Uchino, Yasushi Okuno, Satoaki Matoba.

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
