## [Decision Letter · Decision Letter 0]

2 Sep 2022

PONE-D-22-15581Robust prognostic prediction model developed with integrated biological markers for acute myocardial infarctionPLOS ONE

Dear Dr. Nishi,

Thank you for submitting your manuscript to PLOS ONE. After careful consideration, we feel that it has merit but does not fully meet PLOS ONE’s publication criteria as it currently stands. Therefore, we invite you to submit a revised version of the manuscript that addresses the points raised during the review process. Please respond all raised questions.

We look forward to receiving your revised manuscript.

Kind regards,

Otavio R. Coelho-Filho, M.D., Ph.D., M.P.H.

Academic Editor

PLOS ONE

Journal Requirements:

“Satoaki Matoba was supported by the JST COI Grant Number JPMJCE1302.”              

4. One of the noted authors is a group or consortium “AMI-Kyoto Multi-Center Risk Study Group”. In addition to naming the author group, please list the individual authors and affiliations within this group in the acknowledgments section of your manuscript. Please also indicate clearly a lead author for this group along with a contact email address.

Additional Editor Comments:

Please provide more details about the calibration of the generated model.

The lack of external validation may limit the model's applicability. Please comment and perhaps include it as a potential limitation.

Reviewers' comments:

Reviewer's Responses to Questions

**Comments to the Author**

1. Is the manuscript technically sound, and do the data support the conclusions?

Reviewer #1: Yes

Reviewer #2: Yes

2. Has the statistical analysis been performed appropriately and rigorously? 

Reviewer #1: Yes

Reviewer #2: Yes

3. Have the authors made all data underlying the findings in their manuscript fully available?

Reviewer #1: Yes

Reviewer #2: Yes

4. Is the manuscript presented in an intelligible fashion and written in standard English?

Reviewer #1: Yes

Reviewer #2: Yes

5. Review Comments to the Author

Reviewer #1: In this manuscript, the authors presented the results of a machine learning tool dedicated to the detection of STEMI patients at greater risk of death. The algorithm considered the AMI-Kyoto Multi-Center Risk Study, a multicenter national cohort of acute myocardial infarction held in Japan, which enrolled consecutive acute myocardial infarction patients who were admitted at 23 different hospitals across their country. Based on a well-designed random forest classifier model, the authors found an overwhelming accuracy for in-hospital mortality risk prediction, which overcame the accuracy of traditional risk scores, namely GRACE and TIMI. I recommend this paper should be considered for publication, as it may contribute to the application of ML tools dedicated to risk stratification of STEMI.

Reviewer #2: Please, read attached Word file. The authors stated that their data are available upon reasonable request. Since they were retrieved from electronic health records, I understand that some restrictions may apply.

6. PLOS authors have the option to publish the peer review history of their article (what does this mean?). If published, this will include your full peer review and any attached files.

Reviewer #1: No

Reviewer #2: No

---

## [Author Response · Author response to Decision Letter 0]

21 Sep 2022

Responses to all the reviewers are within 'Response to Reviewers.docx'.

---

## [Editor Report · Decision Letter 1]

24 Oct 2022

Robust prognostic prediction model developed with integrated biological markers for acute myocardial infarction

PONE-D-22-15581R1

Dear Dr. Nishi,

We’re pleased to inform you that your manuscript has been judged scientifically suitable for publication and will be formally accepted for publication once it meets all outstanding technical requirements.

Kind regards,

Otavio R. Coelho-Filho, M.D., Ph.D., M.P.H.

Academic Editor

PLOS ONE

Additional Editor Comments (optional):

The authors present a revised manuscript incorporating all reviewers' suggestions.
---

## [Editor Report · Acceptance letter]

26 Oct 2022

PONE-D-22-15581R1 

Robust prognostic prediction model developed with integrated biological markers for acute myocardial infarction 

Dear Dr. Nishi:

I'm pleased to inform you that your manuscript has been deemed suitable for publication in PLOS ONE. Congratulations! Your manuscript is now with our production department. 

Kind regards, 

on behalf of

Dr. Otavio R. Coelho-Filho 

Academic Editor

PLOS ONE